# Latitudinal Gradient in Urban Pressure and Socio-Environmental Quality: The "Peninsula Effect" in Italy

**Bernardino Romano *, Lorena Fiorini, Chiara Di Dato and Vanessa Tomei**

Department of Civil and Environmental Engineering, University of L'Aquila, 67100 L'Aquila, Italy;
lorena.fiorini@univaq.it (L.F.); chiara.didato@graduate.univaq.it (C.D.D.); vanessatomei@gmail.com (V.T.)
* Correspondence: bernardino.romano@univaq.it; Tel.: +39-862434113

**Abstract:** The purpose of this work is to synthesize, for an international audience, certain fundamental elements that characterize the Italian peninsular territory, through the use of a biogeographical model known as the "peninsula effect" (PE). Just as biodiversity in peninsulas tends to change, diverging from the continental margin, so do some socio-economic and behavioral characteristics, for which it is possible to detect a progressive and indisputable variation depending on the distance from the continental mass. Through the use of 14 indicators, a survey was conducted on the peninsular sensitivity (which in Italy is also latitudinal) of as many phenomena. It obtained confirmation results for some of them, well known as problematic for the country, but contradictory results for others, such as those related to urban development. In the final part, the work raises a series of questions, also showing how peninsular Italy, and in particular Central–Southern Italy, is not penalized so dramatically by its geography and morphology as many political and scientific opinions suggest. The result is a very ambiguous image of Italy, in which the country appears undoubtedly uniform in some aspects, while the PE is very evident in others; it is probably still necessary to investigate, without relying on simplistic and misleading equations, the profound reasons for some phenomena that could be at the basis of less ephemeral rebalancing policies than those practiced in the past.

**Keywords:** peninsula effect; urban pressure; south Italy; latitudinal gradient

---

## 1. Introduction

This work is inspired by the biogeographical phenomenon known as the "peninsula effect" (PE) to analyze and verify the influence of peninsular geography in Italy on various kinds of anthropic activities, through a set of indicators aimed at highlighting environmental, social, and economic aspects of this continental prominence that stretches for about 900 km in the Mediterranean Sea. These same aspects are then investigated in parallel with the urban transformation of the soil which represents, in Italy, one of the most serious pathologies affecting the quality of social life and biodiversity.

The peninsula effect is a classic biogeographical concept which predicts that the number of species declines from a peninsula's basis to its tip. It is an extension of island theory, the subject of experiments in multiple worldwide cases [1–6], on the basis of which there are significant differences in the distribution and quantity/variety of biotic species along the peninsular area [7]. Two of the three hypothesized causal mechanisms—the effects of geological history or habitat on species richness—can be controlled for via the study design and/or statistical analysis. The third proposed mechanism (reduced colonization towards the peninsular tip) is attributed to peninsular geometry and is less easily controlled [8].

The role of the PE can be at the basis of the well-known "southern question"; that is, the overall socio-economic weakness of the south of the country, caused by the stratification of multiple historical

events [9–11], but which in the negative evolution of the last half century probably owes a lot to geographic physiognomy and morphology. In this sense, some considerations can also be trivial, such as the extension of land lines of communication to national and European economic gangs, the objective inefficiency of maritime transport (historically much more important than today), and the design complexity of non-coastal communications, due to the geomorphological harshness of the internal areas, especially in the section of the Central Apennines. However, as is discussed in this paper, many contradictions can be associated with this profile consolidated in common thought, which sees the dynamic of urban transformation as an element capable of provoking questions and reversing convictions. The result is a very ambiguous image of Italy, in which the country appears undoubtedly uniform in some aspects, while the PE is very evident in others; it is probably still necessary to investigate, without relying on simplistic and misleading equations, the profound reasons for some phenomena that could be at the basis of less ephemeral rebalancing policies than those practiced in the past.

## 2. Data and Methods

The measurement of the peninsula effect on various phenomena was carried out using a set of 14 indicators with values calculated on a regional basis. These indicators are derived from the initial idea of comparing the differences along the peninsular arch that concern physical and social aspects to bring out significant differences/homogeneities or links/contradictions. Therefore, research was carried out on the data available at the same level of detail and recently updated for the whole national territory in the demographic, urban, socio-economic, and environmental sectors, and the possibility arose to fill in the 14 indicators used.

For most of the indicators, the source used was ISTAT (Central Institute of Statistics), which systematically publishes data relating to the population and the related economic and social components at least every 10 years (but in many cases also more frequently). The data concerning quality and environmental protection were instead derived from documents of the Ministry of the Environment or, in the case of forests, from the European CORINE Land Cover 2012. Data from the urbanized areas came from the processing carried out by the University of L'Aquila for the 1950s [12,13], while data for the current period came from the digital land use maps developed by the regions until 2008 (LUM).

More up-to-date and efficient data have been produced from 2015 onwards by the National Environmental Research Institute (ISPRA) [14]. Monitoring occurs through the production of a national land use map on a raster basis (regular grid), using data from Copernicus and, in particular, the Sentinel-2 mission [15,16] launched in June 2015, which provides multispectral data with a 10-meter resolution, suitable for both photo-interpretation and semi-automatic classification processes.

For this study, we chose to use regional LUM data and not the ISPRA data updated in 2017, since the latter also include surfaces covered by some categories of interurban roads (not separable) and therefore cannot be compared with the 1950s data that did not include these elements. The urbanized areas surveyed by ISPRA in 2017 exceed those of regional LUMs by approximately 260,000 ha. This can be ascribed, in part, to the increase that has occurred over the past 10 years, but also largely to the inclusion of the road network, amounting to almost 200,000 km out of the approximately 870,000 total in Italy comprising all categories. It must be taken into account that the aspect of urbanization is one of the most complex, as it presents significant typological differences, both within the Italian territory and at the scale of the Mediterranean area [17–19]. In the case of the present research, the phenomenon has been simplified and reduced to the four indicators defined to highlight the more macroscopic characters.

The 14 indicators have been grouped into 5 categories: demographic (demographic density, demographic variation rate, old age index), economic (average income, GDP per capita), social (university degree, unemployment rate), environmental (protected area rate, Natura 2000 rate, forest rate), and urban (urban density, urbanization per capita, urbanization variation rate, land take speed), and their calculation was based on updated data in a chronological range between 2006 and 2011, whose formulations and sources are indicated in Table 1.

**Table 1.** Indicator set.

| | | | **Demographic indicators** | | |
|---|---|---|---|---|---|
| Dd | Demographic density | $Dd = \frac{p}{Ra}$ | p= number of resident inhabitants<br>Ra = Regional area | inhab/km$^2$ | 2011 |
| Dvr | Demographic variation rate | $D_{vr} = \frac{p_{(2011)} - p_{(1950)}}{p_{(1950)}}$ | p (2011)= population on 2011 (number of resident inhabitants)<br>p(1950)= population on 1950 (number of resident inhabitants) | % | 1950–2011 |
| OAi | Old Age Index | $OAi = \frac{p_{>65}}{p_{<15}}$ | p>65 = population older than 65 years<br>p<15 = population younger than 15 years | % | 2011 |
| | | | **Economic indicators** | | |
| AI | Average income | $AI = \frac{Rcr}{p_{(2011)}}$ | Rcr = Total regional income<br>p(2011)= number of resident inhabitants on 2011 | €/inhab | 2011 |
| GDPpc | GDP per capita | $GDPpc = \frac{GDPr}{p_{(2011)}}$ | GDPr= Total regional GDP<br>p(2011)= number of resident inhabitants on 2011 | €/inhab | 2011 |
| | | | **Social indicators** | | |
| Ud | University degree | $Ud = \frac{p_u}{p_t}$ | pu = population between 30–34 years old with university degree<br>pt = total regional population between 30 and 34 years old | % | 2011 |
| Ur | Unemployment rate | $Ur = \frac{p_{un}}{p_{ar}}$ | pn = regional population unemployment<br>par = total regional population | % | 2011 |

**Table 1.** *Cont.*

| | | | **Environmental indicators** | | |
|---|---|---|---|---|---|
| Par | Protected areas rate | $PAr = \frac{PAa}{Ra}$ | PAa = Regional protected areas area <br> Ra = Regional area | % | 2018 |
| N2kr | Natura 2000 rate | $N2kr = \frac{N2ka}{Ra}$ | N2ka = regional N2k area <br> Ra = regional area | % | 2018 |
| Fr | Forest rate | $Fr = \frac{Fa}{Ra}$ | Fa = regional forest areas <br> Ra = regional area | % | CORINE 2018 |
| | | | **Urbanization indicators** | | |
| Ud | Urbanization density | $Ud = \frac{Ua}{Ra}$ | Ua = urbanized areas <br> Ra = regional area | $m^2/km^2$ | 2008 |
| Upc | Urbanization per capita | $Ud = \frac{Ua}{n_{inhabit.}}$ | Ua = urbanized areas <br> N(inhab) = resident inhabitants | $m^2/ab$ | 2008 |
| Uvr | Urbanization variation rate | $U_{vr} = \frac{Ua_{(2008)} - Ua_{(1958)}}{Ua_{(1950)}}$ | Ua = urbanized areas | % | 1958–2008 |
| LTs | Land take speed | $LT_s = \frac{Ua_{(2008)} - Ua_{(1958)}}{ndays}$ | Ua = urbanized areas <br> ndays = Number of days in considered time range (50 years) | ha/day | 1958–2008 |

The values assumed by the various parameters were then sorted by latitudinal succession from north to south according to the geographical location of the regions deduced from the position of their centroids (Figure 1). The ordinal positioning of the data made it possible to verify the degree of sensitivity of the phenomena correlated with latitude; and therefore, trend curves of order 2 and the relative values of the determination coefficients $R^2$ were elaborated as a square of the Pearson coefficient which, varying between 0 and 1, is a proportion between the variability of the data and the correctness of the statistical model used. The higher values of $R^2$ therefore denounce a greater sensitivity of the phenomena analyzed to the latitudinal factor and, thus, a more marked affirmation of the EP. A further parameter analyzed is the standard deviation on the average (SD), which returns, at the variation of its value, the concentration/dispersion of the data with respect to the average of the same and therefore measures the homogeneity of behavior of the regions towards the single phenomenon considered.

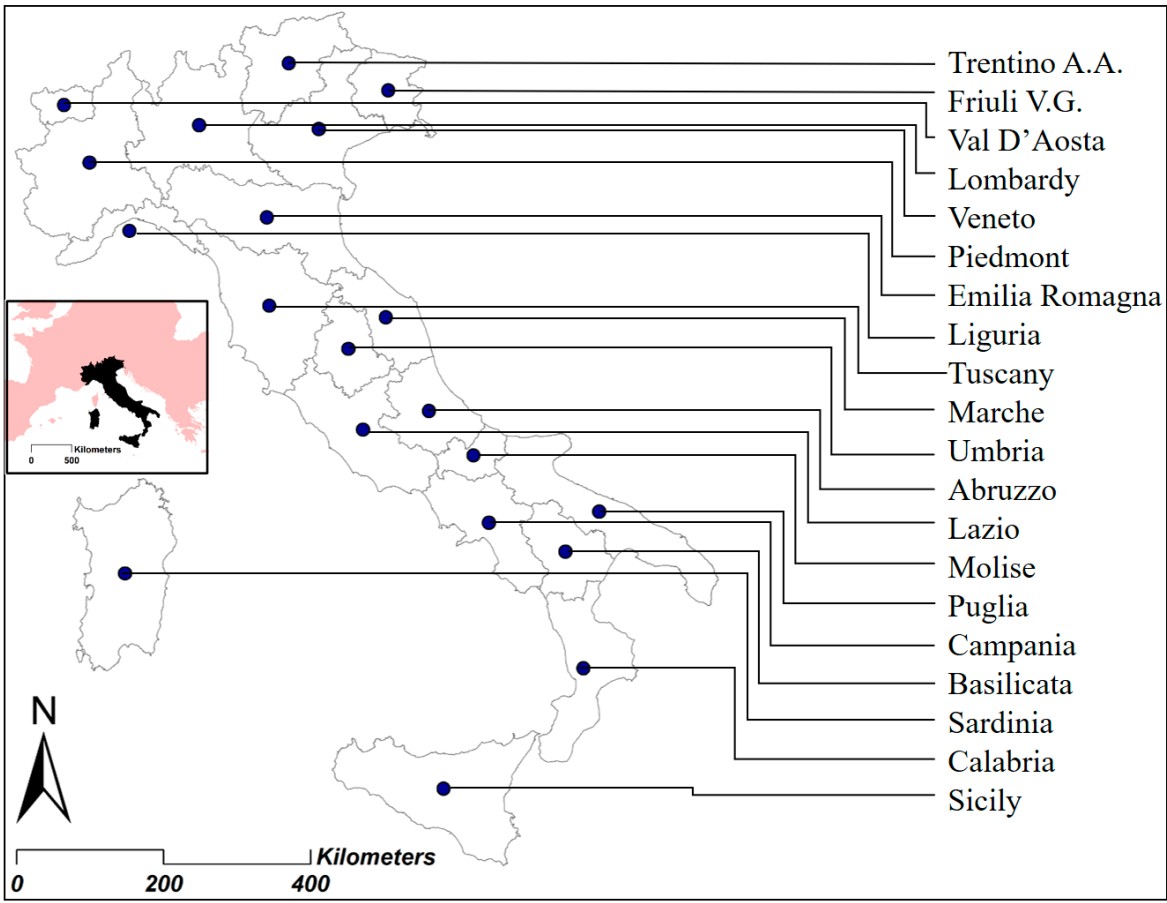

**Figure 1.** Latitudinal gradient of Italian regions

## 3. Results

### 3.1. Analytical Outcome

The latitude sensibility survey shows a phenomenological response fairly aligned by categories (Figure 2), while Figure 3 highlights the complementarity between the coefficient $R^2$ and SD. The demographic indicators show a significant indifference with respect to the PE of the regions, considering that the Dd and the DVR have very low $R^2$ with a high degree of inhomogeneity (highlighted by the SD compared to the average). The most homogeneous parameter by region (SD = 0.18) is the OAI index which, with an $R^2$ higher than the other two, testifies to a substantial uniformity of the Italian regions with respect to the aging population. The most marked differences, associated with a very clear latitudinal sensitivity, concern the economic and social indicators. In this case (at least for AI,

GDPpc, and Ur), there is an important approach of the $R^2$ coefficients to the maximum value with a variation in the order of magnitude in comparison with the demographic ones (all close to or above 0.7) with a geometry indisputably governed by the latitudinal gradient. The least contaminated parameter in this sense is the level of university education (Udr) of the population aged between 30 and 34, which fluctuates relatively little between regions (SD = 0.16) and is recorded by the trend curve in fairly reliable form ($R^2$ = 0.53); there is a slight predominance of the regions of Central Italy, but with a distribution that is not particularly penalizing towards the south of the country. The PE does not manifest itself in a striking form when considering the distribution of the areas of environmental value (Par, N2kr, Fr): these areas, which are important because they host much of the national ecological network [20], have fairly high SD values (above 0.4), which denote a high dispersion of conditions; the $R^2$ values remain very low (less than 0.08) and the latitudinal dependence appears so slight that it cannot be considered statistically significant. The category with the greatest internal differences is that of urban planning indicators, for which latitudinal sensitivity is often disjointed and therefore dominated by more local dynamics. The SD values produce a scenario of behavioral independence, at least for the Ud, Upc, and Uvr (urbanization variation rate) indicators (SD higher than 0.3), while the LTs parameter appears less dispersed (SD = 0.04). The greater visibility of the latitudinal gradient regards the Uvr parameter which, with an $R^2$ = 0.35, shows a center–south which, in the last half century, has seen its urbanized areas grow proportionately well more than the northern regions, acquiring per capita urbanization levels (Upc) which are fully comparable and certified on the national average of approximately 370 m$^2$/inhab. On the other hand, the other indicator of settlement behavior (the first is precisely the Upc), or the LTs (land take speed), which shows a huge misalignment of the regions even if a prevalence is distilled in a group of central and northern regions, is difficult to refer to a homogeneous behavior.

It is quite interesting to evaluate how aligned behaviors on the urban growth and higher education front do not have equally aligned consequences on the economic and social front. Evidently, the entrepreneurial advantages coming from the construction industry have not remained localized in the center–south, where they have only produced negative environmental effects and temporary and secondary economic returns.

In summary, the functions that derive from the latitudinal processing of the 14 indicators analyzed are aggregated into 4 clusters of types (Figure 4):

1. a low PE on 2 environmental indicators (Fr and N2kr) and urban dynamics (DVR and LTs);

2. a slight phenomenological prevalence in the central peninsula for three social indicators (Dd, Ud, and OAI);

3. a net incremental PE from north to south for an environmental indicator (Par), a social one (Ur), and an urban one (Uvr);

4. a net decreasing PE from north to south for two economic indicators (AI and GDPpc) and an urban one (Upc).

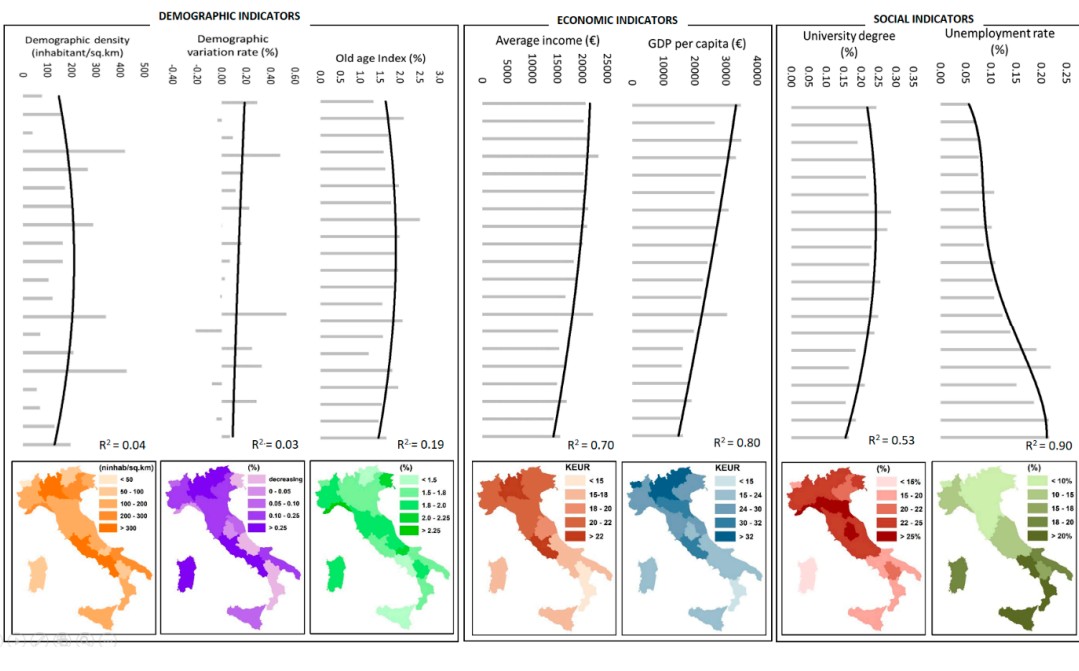

**(a).**

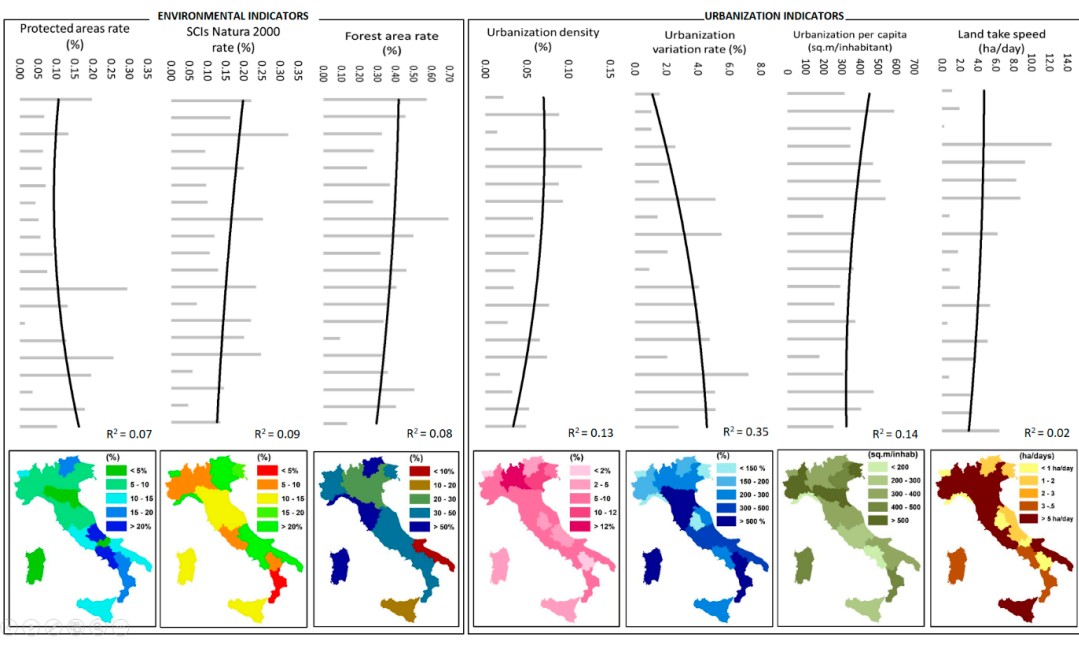

**(b).**

**Figure 2.** Indicators of latitude sensibility. (**a,b**)In the figures there are the plots showing the trend of the indicator values along the latitudinal axis of the peninsula with the maps showing the spatial distribution by region of the same indicators classified by range.

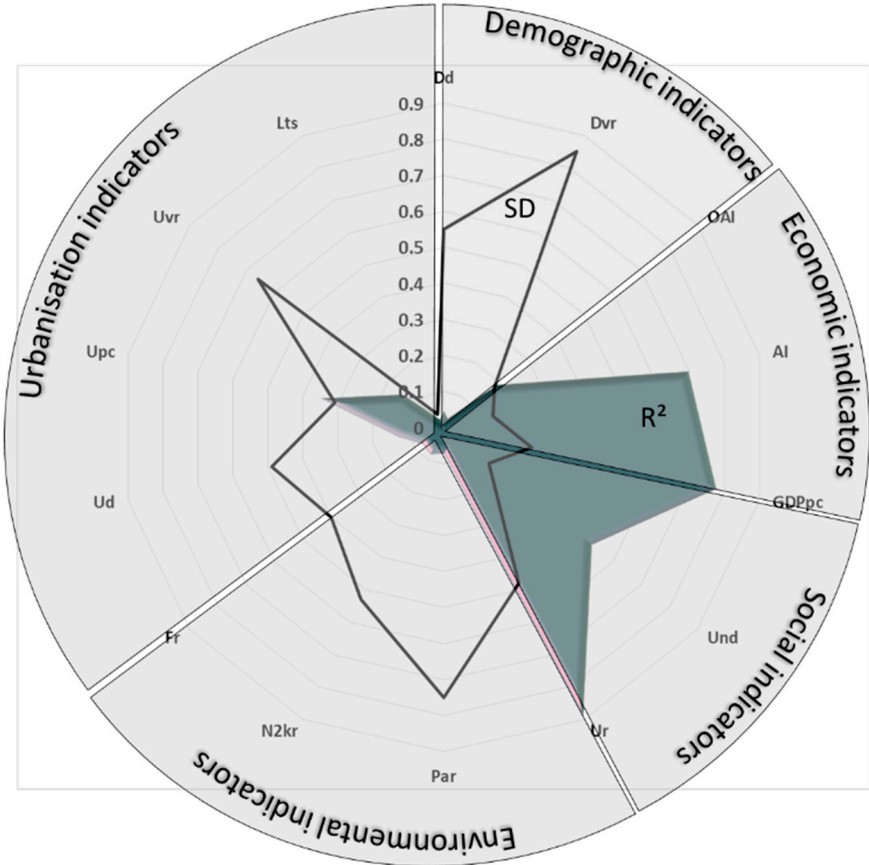

**Figure 3.** Relationship between standard deviation (SD) and $R^2$ parameters.

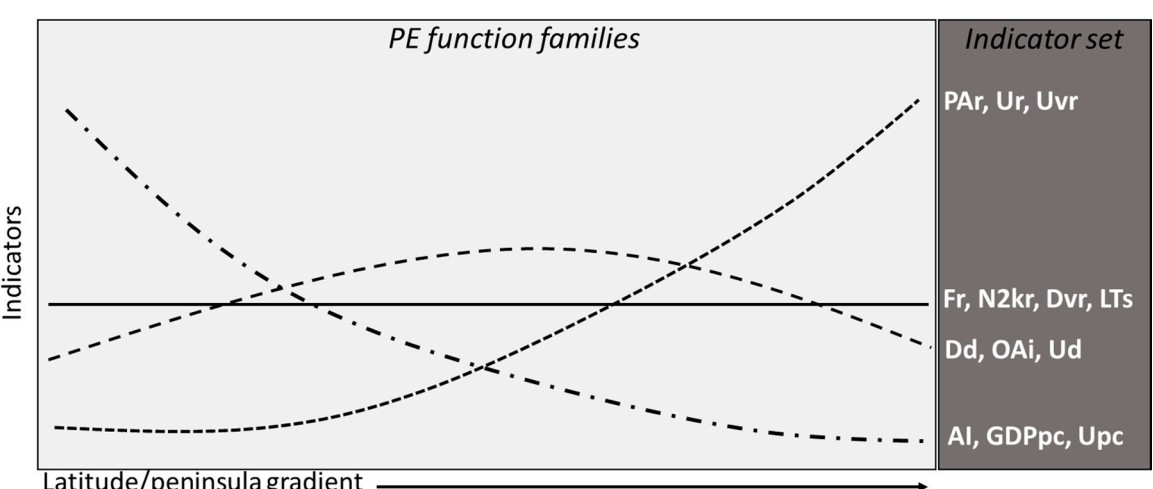

**Figure 4.** Peninsula effect (PE) function cluster.

### 3.2. Comparative Assessment

Italy is actually much more uniform along the latitudinal axis than the dominant and widespread thought expresses; or at least it is in many important aspects, such as environmental quality and biodiversity, demographic variations, population structure, and degree of education. In the face of these components, which are not extremely variable between the geographical positions of the regions, there are the social and economic ones, which show such a clear gradient as to be indisputable [21] which, for the same other resources, is rather contradictory. Many explanations that are formulated

in the political and media fields apply too simplistic equations, one of the most typical of which is linked to the action of organized crime in economic management [22]. However, this cannot be the reason alone behind such an accentuated economic weakness; in recent decades, organized crime has spread widely throughout the peninsula and in particular in its richer areas. As was mentioned at the beginning of this paper, a reasonable cause could lie in the length and inefficiency of the land connection lines, but even this factor cannot be so decisive, as it is true that mobility in the south is more difficult [23], but the entire peninsular length is affected by motorway lines, and the transport of goods by road is very intense every day throughout the year. The peninsula, extended for about 900 km in the Mediterranean, is innervated by more than 600 km of TAV (high-speed railway), 2000 km of inter-city railways, about 2500 km of motorways and 26 airports, of which at least 9 have multi-day communications with the regions of Northern Italy (Figure 5).

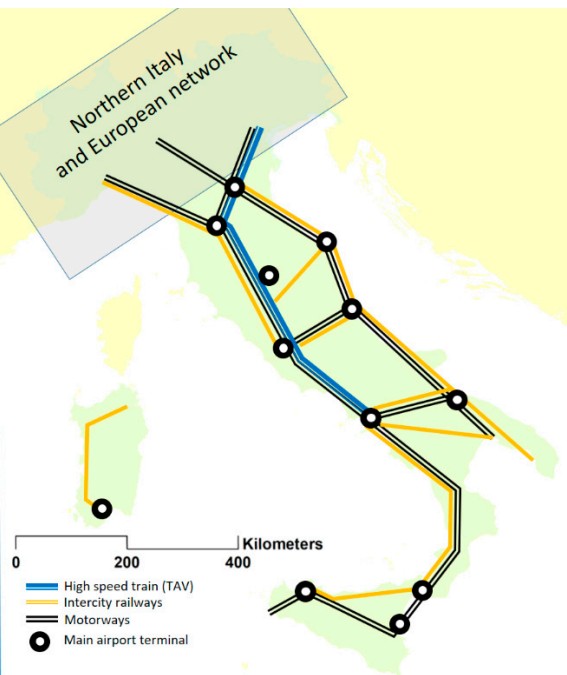

**Figure 5.** Main infrastructural peninsula equipment.

The TAV railway lines arrive on the Tyrrhenian coast up to Salerno; that is, covering almost 4/5 of the peninsular development and more than half of the geographical area defined as Central and Southern Italy, while the opposite Adriatic coast, despite not having the TAV line, has been equipped with inter-city trains for many years, which ensure connections throughout the development several times a day. The exchanges of interest with the northern regions are continuous and intense: just think of summer tourism that sees real exoduses from the north to the south for seaside activities, with a large spread of second homes in all the Tyrrhenian–Adriatic–Ionian coastal areas and major and minor islands. There is no doubt that southern society manifests behavioral inefficiencies in public–private management [24] and in territorial planning. The PE that we talked about in the article re-emerges quite categorically in the survey conducted on the provision and updating of the urban planning tools of the municipalities (Figure 6); but in truth with many exceptions concerning Sardinia, Sicily, and Puglia which place them at a comparable level with some northern regions, such as Friuli or Piedmont and Liguria [25], which hardly anyone in Italy would say are lagging behind on the side of their municipal planning.

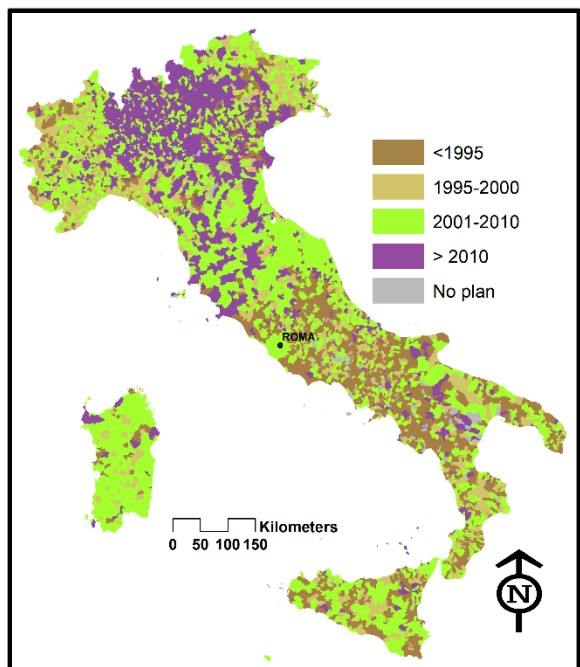

**Figure 6.** Geographical distribution of plan update thresholds (credit: re-processing of National Institute of Urban Planning-INU, 2016 data).

## 4. Conclusions

The survey carried out and presented in the work is aimed at highlighting the macroscopic contradictions that the Italian peninsula manifests towards some socio-economic, environmental, and settlement phenomena which, in normal communication, are generally exposed with excessive simplification. Certainly, a central aspect that then becomes a driving force for reflection for all the others is the great energy of urban transformation that has affected this geographical area in the last 50 years, without significant differences with the continental national areas. Indeed, some indicators such as the Uvr (urbanization variation rate) show an incremental curve markedly accentuated with decreasing latitude (Figure 2). However, this remarkable construction and urban planning activity, which has lasted for more than half a century at a speed completely comparable with that of Northern Italy, has failed to cause stable economic consequences. Other analogies concern the alignment of the country, a peninsular and continental part, with respect to the seniority of the population, the university education rate, and the provision of naturalistic–environmental values, but all this does not prevent the economic and employment quality from being, dramatically and with rigorous statistical reliability, unbalanced towards the south with progressive latitudinal variation. This work does not pretend to provide answers to this complex condition of inconsistency, which, however, has worsened more and more over the years, despite the governments' attempts to compensate and stem it [26,27], but it also wants to point out, also using the other supporting considerations (Figures 5 and 6), how it is necessary to activate more incisive research and intervention channels capable of reading phenomena with different optics from the past.

**Author Contributions:** All authors have read and agree to the published version of the manuscript. Conceptualization, B.R. and L.F.; methodology, L.F.; data curation, C.D.D. and V.T.; writing—original draft preparation, B.R. and L.F.; writing—review and editing, L.F.; supervision, B.R.

**Funding:** This research was funded by 2019 RIA Project (Research of L'Aquila University Interest).

**Acknowledgments:** We thank Prof. Alessandro Marucci and Prof. Francesco Zullo for suggestions and indications relating to the retrieval of data used for research, and the anonymous reviewers who, with their comments, have allowed us to significantly improve the quality of the paper.

**Conflicts of Interest:** The authors declare no conflict of interest.

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
