# Peer review of "Latitudinal Gradient in Urban Pressure and Socio-Environmental Quality: The “Peninsula Effect” in Italy"

_land, doi:10.3390/land9040126_

Round 1
Reviewer 1 Report
It is an interesting and novel paper. The analysis is quantitative and this makes the Discussion of the Conclusions interesting. The Abstract provides a representative summary of the paper and the paper's title is well chosen. The paper “ Latitudinal Gradient in Urban Pressure and Socio-Environmental Quality: The “Peninsula Effect” in Italy” would be appealing for the audience of Land Journal.
The study synthesizes, fundamental variables that characterizes the Italian peninsular territory, through the use of “peninsula effect” model. According to the analysis of 14 indicators (demographic, economic, social, environmental, and urban variables) on the peninsular sensitivity of many phenomena,they ordain confirmation results for some of them, but contradictory for others, such as those related to urban development.
There are some minor issues that the authors could consider and possibly address.
[1] Figure 2. Indicators of latitude sensibility.
Needs re-drawing in a bigger scale so that it can be read
[2] Probably, at the present time, still in full expansion of the Covid-19 emergency, no final information can be drawn οn the effects of the pandemic. So I consider to delete any reference you make to the epidemic (complex, multilevel and different issue). There are other important issues that can be raised justifying the significance of the results focusing mostly οn urban development.
[3] I consider that the urbanization parameter is not sufficiently documented. In order to succeed a more integrated and international understanding it is worth mentioning briefly the differences of urban characteristics of European cities as wells as, referring to examples of other national cases, concerning the way of urban development namely, expanding cities and the urban sprawl. Possibly this needs to be discussed further in a short paragraph. For example refer to the works of:
- Stathakis D., Tsilimigkas G., 2015. “Measuring the compactness of European medium-sized cities by spatial metrics based on fused data sets”. International Journal of Image and Data Fusion, Volume 6, Issue 1/2015, Pages: 42-64. (DOI: 10.1080/19479832.2014.941018)
- Chorianopoulos, I. Pagonis, T. Koukoulas, S. Drymoniti, S. (2010) Planning, competitiveness and sprawl in the Mediterranean city: The case of Athens. Cities 27(4): 249-259.
- Lagarias, A., Sayas, J. 2018. “Urban sprawl in the mediterranean: Evidence from coastal medium-sized cities”. Regional Science Inquiry. 10(3), pp. 15-32
Author Response
Reviewer 1
It is an interesting and novel paper. The analysis is quantitative and this makes the Discussion of the Conclusions interesting. The Abstract provides a representative summary of the paper and the paper's title is well chosen. The paper “ Latitudinal Gradient in Urban Pressure and Socio-Environmental Quality: The “Peninsula Effect” in Italy” would be appealing for the audience of Land Journal.
The study synthesizes, fundamental variables that characterizes the Italian peninsular territory, through the use of “peninsula effect” model. According to the analysis of 14 indicators (demographic, economic, social, environmental, and urban variables) on the peninsular sensitivity of many phenomena,they ordain confirmation results for some of them, but contradictory for others, such as those related to urban development.
There are some minor issues that the authors could consider and possibly address.
[1] Figure 2. Indicators of latitude sensibility. Needs re-drawing in a bigger scale so that it can be read
The legends of the maps have been redesigned and the characters of the diagrams have been enlarged to be more readable
[2] Probably, at the present time, still in full expansion of the Covid-19 emergency, no final information can be drawn οn the effects of the pandemic. So I consider to delete any reference you make to the epidemic (complex, multilevel and different issue). There are other important issues that can be raised justifying the significance of the results focusing mostly οn urban development.
References to COVID-19 have been removed
[3] I consider that the urbanization parameter is not sufficiently documented. In order to succeed a more integrated and international understanding it is worth mentioning briefly the differences of urban characteristics of European cities as wells as, referring to examples of other national cases, concerning the way of urban development namely, expanding cities and the urban sprawl. Possibly this needs to be discussed further in a short paragraph. For example refer to the works of:
Stathakis D., Tsilimigkas G., 2015. “Measuring the compactness of European medium-sized cities by spatial metrics based on fused data sets”. International Journal of Image and Data Fusion 6(1): 42-64. DOI: 10.1080/19479832.2014.941018
Chorianopoulos, I. Pagonis, T. Koukoulas, S. Drymoniti, S., 2010. Planning, competitiveness and sprawl in the Mediterranean city: The case of Athens. Cities 27(4): 249-259. DOI: 10.1016/j.cities.2009.12.011
Lagarias, A., Sayas, J. 2018. Urban sprawl in the mediterranean: Evidence from coastal medium-sized cities. Regional Science Inquiry. 10(3):15-32
Has been add the following text and the suggested references:
It must be taken into account that the aspect of urbanization is one of the most complex, as it presents significant typological differences, both within the Italian territory and at the scale of the Mediterranean area [15,17]. In the case of the present research, the phenomenon has been simplified and reduced to the four indicators defined to highlight the more macroscopic characters.
Reviewer 2 Report
I have reviewed the revised paper entitled “Latitudinal Gradient in Urban Pressure and Socio-Environmental Quality: The Peninsula Effect in Italy” (land-771460). The authors adopt 14 indicators to conduct the peninsular and latitudinal sensitivities. However, the authors explain the background of 14 indicators unclearly. It is still not clear that why the authors use these 14 indicators or why the authors “only” use these 14 indicators. The authors should explain the reasons very clearly, otherwise the contribution of the entire paper will be questioned. On page 2 and 3, the authors mention the meaning of value of R2, but ignore to address the meaning of value of SD. Figure 2 is too vague to be understood. The economic indicators and social indicators seem to be the main effects on the peninsula effect in Italy in Figure 3. In discussion section, the authors mention the geographic distribution of Covid-19 positive cases and it is quite interesting. However, the Covid-19 transmission route is more complicated (for example, through the public transportation system, sightseeing, etc.), I am afraid that it cannot be discussed with only 14 indicators.
Author Response
I have reviewed the revised paper entitled “Latitudinal Gradient in Urban Pressure and Socio-Environmental Quality: The Peninsula Effect in Italy” (land-771460). The authors adopt 14 indicators to conduct the peninsular and latitudinal sensitivities.
However, the authors explain the background of 14 indicators unclearly.
It is still not clear that why the authors use these 14 indicators or why the authors “only” use these 14 indicators. The authors should explain the reasons very clearly, otherwise the contribution of the entire paper will be questioned.
Has been add the following text:
These indicators are derived from the initial idea of comparing the differences along the peninsular arch that concerned physical and social aspects to bring out significant differences/homogeneities or links/ contradictions. Therefore, a research was carried out on the data available at the same level of detail and recent updating for the whole national territory in the demographic, urban, socio-economic and environmental sectors and the possibility arose to fill in the 14 indicators used.
On page 2 and 3, the authors mention the meaning of value of R2, but ignore to address the meaning of value of SD.
Has been precised that:
A further parameter analyzed is the standard deviation on the average (SD) which returns, at the variation of its value, the concentration/dispersion of the data with respect to the average of the same and therefore measures the homogeneity of behavior of the regions towards the single phenomenon considered
Figure 2 is too vague to be understood.
The legends of the maps have been redesigned and the characters of the diagrams have been enlarged to be more readable
The economic indicators and social indicators seem to be the main effects on the peninsula effect in Italy in Figure 3.
Has been precised that:
The most marked differences, associated with a very clear latitudinal sensitivity, concern the economic and social indicators. In this case (at least for AI, GDPpc, and Ur), there is an important approach of the R2 coefficients to the maximum value with a variation in the order of magnitude in comparison with the demographic ones (all close to or above 0.7) with a geometry indisputably governed by the latitudinal gradient.
In discussion section, the authors mention the geographic distribution of Covid-19 positive cases and it is quite interesting. However, the Covid-19 transmission route is more complicated (for example, through the public transportation system, sightseeing, etc.), I am afraid that it cannot be discussed with only 14 indicators.
Text and references to COVID-19 have been removed
This manuscript is a resubmission of an earlier submission. The following is a list of the peer review reports and author responses from that submission.